# Community Solutions to Increase the Healthfulness of Grocery Stores: Perspectives of Immigrant Parents

**DOI:** 10.3390/ijerph20156536

**Published:** 2023-08-07

**Authors:** Hadis Dastgerdizad, Rachael D. Dombrowski, Bree Bode, Kathryn A. G. Knoff, Noel Kulik, James Mallare, Ravneet Kaur, Heather Dillaway

**Affiliations:** 1Department of Public Health, University of South Carolina, Bluffton, SC 29909, USA; 2Departments of Public Health and Kinesiology, College of Education, Health and Human Services, California State University-San Marcos, San Marcos, CA 92096, USA; rdombrowski@csusm.edu; 3Michigan Fitness Foundation, Lansing, MI 48314, USA; bbode@michiganfitness.org; 4Office of Policy Support, Food and Nutrition Service, US Department of Agriculture, Alexandria, VA 22314, USA; kathryn.knoff@usda.gov; 5Center for Health and Community Impact, Division of Kinesiology, Health & Sport Studies, College of Education, Wayne State University, Detroit, MI 48202, USA; ab7564@wayne.edu; 6Department of Family Medicine and Public Health Sciences, Wayne State University, Detroit, MI 48202, USA; jpmallare@wayne.edu; 7Division of Health Research and Evaluation, Department of Family and Community Medicine, College of Medicine, University of Illinois, Rockford, IL 61107, USA; ravneetk@uic.edu; 8Department of Sociology and Anthropology, Illinois State University, Normal, IL 61790, USA; hedilla@ilstu.edu

**Keywords:** qualitative multiple-case study, acculturation, independently owned grocery stores, immigrant parents, early childhood obesity, targeted marketing, sugar sweetened beverages

## Abstract

Grocery store environments are recognized as one of the most crucial community settings for developing and maintaining healthy nutritional behaviors in children. This is especially true for disadvantaged ethnic minority families, such as immigrants, who reside in the Detroit Metropolitan area and have historically experienced inequities that result in poor health outcomes. Rates of obesity and type II diabetes have affected Detroit 38% more than the rest of the state and nationwide. In 2019, almost 54% of children aged 0–17 in Metro Detroit lived in poverty, and 21.6% experienced food insecurity, compared with the state level of 14.2%. Moreover, nearly 50% of ethnic minority children in Metro Detroit consume sports drinks, and 70% consume soda or pop in an average week. The primary purpose of this study was to explore immigrant parents’ perspectives on (1) how in-store Sugar-Sweetened Beverage (SSB) marketing impacts the purchasing behaviors of parents and the eating behaviors of toddlers, and the secondary objective was to (2) determine strategies to reduce SSB purchases and consumption within grocery environments from the viewpoints of immigrant parents. A qualitative multiple-case study design was used to achieve the aims of this study. Semi-structured individual interviews were completed with 18 immigrant parents of children aged 2 to 5 years old who were consumers in 30 independently owned full-service grocery stores within the immigrant enclaves of Detroit, Dearborn, Hamtramck, and Warren, Michigan. Three key thematic categories emerged from the parents’ narratives. These themes were: (1) non-supportive grocery store environments; (2) acculturation to the American food environment; and (3) strategies to support reduced SSB consumption among young immigrant children. The findings of this study revealed widespread SSB marketing targeting toddlers within the participating independently owned grocery stores. Even if families with young children practiced healthy nutritional behaviors, the prices, placements, and promotion of SSBs were challenges to establishing and sustaining these healthy eating habits. The parents believed that planning and implementing retail-based strategies in collaboration with families and considering families’ actual demands would assist in managing children’s eating patterns and reducing early childhood obesity.

## 1. Introduction

Over the past 20 years, 30% of the world’s population has become obese or overweight [1,2,3]. In the United States (U.S.), the prevalence of obesity in adults and children aged 2–19 has increased since 1980. Between 1980 and 2020, the prevalence of obesity in adults and children and severe obesity in adults increased from 15% to 41.9%, 6% to 19.7%, and 4.7% to 9.2%, respectively [1,2,3]. Childhood obesity is a rising health problem and can cause health consequences, including type II diabetes, anxiety, depression, self-esteem issues, and a lower quality of life [1,4,5,6,7]. If existing obesity rates continue to rise, nearly half of the children in the world will be overweight or obese by 2030, with considerable health consequences [1,8,9].

The high prevalence of overweight and obesity cannot be attributed solely to genetic factors [10,11,12]. Individual behaviors, social and cultural variables, and obesogenic eating environments also play essential roles in the development of obesity [13,14,15]. The early nutritional environments and feeding practices performed by adults around children can shape children’s long-term eating behaviors and skills [16,17]. Grocery stores are usually the primary locations for family food purchases and can be vital in promoting or hindering healthy eating behaviors among young children [18,19,20]. Studies have shown that increasing the number of healthy grocery stores near children’s homes can improve children’s healthy eating habits and weight status and positively impact their Body Mass Index (BMIs) [21,22,23]. However, the retail grocery setting has historically been primed for numerous food and beverage marketing efforts, especially for ultra-processed foods and sugar-sweetened beverages (SSBs), which are recognized obesogenic environmental indicators [18,24,25,26,27]. Food marketing in these settings can also simultaneously impact parents’ attitudes and children’s preferences through stimulant techniques such as pricing, placement, and promotion of products [19,27,28,29]. These marketing efforts often target young children and are linked to high consumption of unhealthy foods and beverages [19,30,31,32,33,34]. Food companies have also tried to make their less-healthy products culturally relevant to effectively target ethnic minority groups, such as immigrant families [34]. For instance, tactics have included choosing Spanish names for products and advertising on Spanish-language television channels or African American television programs that are known to be culturally favored [18,34,35]. Food companies focus their marketing efforts much less on promoting healthier foods and unsweetened beverages to ethnic and immigrant groups [34,35,36,37]. Targeted food and beverage marketing has contributed to increased consumption of SSBs among Black/African Americans and Latinos/Hispanics, which subsequently has incited a higher prevalence of obesity, type II diabetes, and heart disease among these groups [38,39,40,41,42,43,44]. Differences in SSB consumption can even be seen at the ages of six to eleven months among ethnic/minority groups [45,46,47,48]. In 2018, 74% of Latino children and 82% of African American children were found to have consumed some SSBs by the age of two, compared with only 45% of non-Hispanic Whites [45,46,47,48]. High SSB consumption during early childhood increases the likelihood of childhood obesity, which leads to obesity in adulthood [1,43,44,47,48], as has been seen in 25.8% of children of Hispanic descent and 22.0% of African American children. These rates are higher than those in non-Hispanic White children of 14.1% [1,43,44].

Additionally, studies on immigrants indicate that acculturation to processed foods is another socio-environmental factor that contributes to the development of obesity within the immigrant population [38,39]. Study findings have shown that the average BMIs of men and women in Nigeria are 21.7 and 22, and in the U.S., the average BMIs of Nigerian men and women are 27.1 and 30.8, respectively [40]. A similar rate is seen in Pima Indians living in the U.S., who are 25 kg heavier than Pima Indians living in Mexico [41,42]. Nearly 50% of Mexican Americans and 43.2% of Mexican immigrants have high BMIs, compared with only 29.3% of Mexicans residing in Mexico [43,44]. Likewise, immigrant children who live in the U.S. have higher BMIs than their peers in their home countries, regardless of their country of birth [43,44]. Furthermore, lifestyle and behavioral differences between foreign- and US-born Latino/Hispanic immigrants have been shown to contribute to the increase in overweight prevalence between the first and subsequent generations of US residents within these populations [45,46,47,48]. Current reformatory strategies have not been enough to tackle obesogenic food environments, and SSB products are still highly promoted in grocery stores [14]. Therefore, it is important to expand and strengthen efforts to protect children, specifically those with minority backgrounds, from the increasing marketing tactics and eventual consumption of SSBs [5,35,37,49,50,51,52]. This study attempted to broaden the understanding of immigrant parents’ viewpoints on SSB marketing and consumption and how the negative impacts of SSB marketing on children’s nutritional behaviors could be reduced.

Eating environments are important venues for communicating with large population segments to promote health and can be utilized to implement healthy eating interventions [30,53,54,55,56,57]. However, modifying the environmental risk factors for childhood obesity requires multi-level interventions, including parents’ involvement [55]. As primary providers of food and beverages for their children, parents play a significant role in developing and establishing children’s healthy eating behaviors [58]. Likewise, parents’ perceptions of water safety can influence their consumption of water or an alternative, such as SSBs [48,57,59,60,61]. Robust evidence indicates that the perception of water quality is most influenced by belonging to racial or ethnic minority groups, foreign-born groups, and having limited access to safe drinking water [48,59,60,61]. Studies in Michigan (MI) after the Flint Water Crisis have shown that even after tap water was deemed safe, there is still significant mistrust of the local water system and increased preferences for drinking other alternatives, such as bottled water or SSBs, among ethnic minority groups, which could be a reason for a lower water consumption rate in this state [53,57,60,61,62,63]. Additionally, targeted food and beverage marketing can confuse caregivers regarding the benefits of different beverages and foods, which causes many parents to believe that SSBs, including juices and toddler SSB drinks, are healthy options for their children [61,62]. The results of studies have shown that Asian, non-Hispanic Black, and Hispanic/Latino parents believed that non-100% fruit juices and sports drinks were healthy because of their fruit contents and were more likely to serve SSBs to their young children compared with White non-Hispanic parents [57,58,59,60,61,62,63,64]. Moreover, immigrants are often in a social crisis with persistent social instability and a significant burden of chronic diseases [65,66,67,68,69]. High stress levels, usually due to financial problems, acculturation, language barriers, duration of residency, and perceived discrimination-based traumatic stress, have been found to be directly related to an increased prevalence of health issues among Mexican and Arab American immigrants [70]. For instance, limited English language proficiency causes more acculturative stress, negatively influencing an immigrant’s health status [71,72]. Immigrants have different health needs and are more exposed to potentially health-damaging situations, such as cultural shock [73], and most current health promotion interventions targeting non-immigrant populations may not work for immigrants and minority groups [73].

It is important to engage immigrant families in the Metro Detroit area, where multiple factors contribute to health inequities. Immigrant families living in the Metro Detroit area report a high prevalence of chronic diet-related disorders and limited access to healthy eating and retail environments. Metro Detroit’s residents have been impacted by poverty, bankruptcy, unemployment, and high obesity rates. Collectively, these variables impact nutritional behaviors and health outcomes, creating an inequitable environment for children living in Metro Detroit, specifically ethnic minority immigrant children. One way to address these concerns is through the Great Grocer Project (GGP), which is the principal study of the current study and was launched in 2020 [74]. The GGP has implemented healthy food marketing strategies within several independently owned grocery stores in low-income and underserved neighborhoods of Metro Detroit to improve families’ healthy eating behaviors and health outcomes within MI [74,75,76,77]. The current study employed a qualitative multiple-case study approach to identify the key strategies to reduce the damaging effects of SSB marketing on the health of young children in immigrant families. Two aims were explored: (1) to determine how in-store SSB marketing impacts the purchasing behaviors of immigrant parents and the eating behaviors of their young children; and (2) to identify strategies to reduce SSB purchases and consumption from the viewpoints of immigrant parents with young children.

## 2. Materials and Methods

The constructs of the Social Ecological Model and Community Nutrition Environment guided this qualitative study. In this study, to determine strategies to reduce SSB purchases and consumption by immigrant families with young children, it was first necessary to explore the parents’ attitudes and knowledge regarding healthy eating, SSB marketing and consumption, and the challenges they face if they purchase fewer SSBs for their children. Narrative reports clarified this study’s aims and assisted in determining store- and family-based strategies to reduce the purchases and consumption of SSBs from the perspectives of the immigrant parents of children aged 2 to 5 years old. A qualitative multiple-case study approach was used to respond to the research questions. By involving multiple cases and centering the narrative reports, the authors explored patterns of similarity or difference across the lived experiences of the immigrant parents impacted by SSB marketing within independently owned grocery stores, generated broad thematic categories, and attained a clearer understanding of the issues. The data collection included conducting semi-structured one-on-one interviews [76,77]. The semi-structured interviews included four sets of questions, including: (1) immigrant parents’ backgrounds and attitudes toward healthy eating; (2) current in-store SSB marketing tactics; (3) SSB consumption; and (4) the impacts of family culture, religion, and immigration on forming, practicing, and sustaining healthy eating habits. These topic areas were explored throughout the interview process to obtain collective responses to the research questions and aims of this study. The participants’ perspectives and viewpoints were used to understand how grocery environments could be improved while also aiming to reduce early childhood obesity impacted by SSB marketing within independently owned grocery stores.

### 2.1. Setting and Characteristics

The current qualitative multiple-case study targeted consumers in 30 independently owned grocers that were participating stores in the Great Grocer Project. Participant grocery stores in this study were all large-format, full-service grocery stores with an abundant selection of food offerings. Given that the stores are independently owned vs. corporately owned and located in an urban environment, some food products may sell at higher prices in their stores than competitors (e.g., Wal-Mart) [77,78,79,80]. Participating grocers were located in Hispanic and Latino enclaves in Southwest Detroit and Arab and Chaldean enclaves in North Central Detroit (N = 8 stores), Dearborn (N = 13 stores), Hamtramck (N = 3 stores), and Warren, MI (N = 6 stores). These cities represent four cases in this study and are located in two different counties (Wayne and Macomb Counties). The Wayne and Macomb Counties are the most populous and diverse in MI (i.e., regarding socioeconomic status and racial/ethnic minorities) and host nearly 401,100 immigrants, corresponding to 63% of the state’s foreign-born population [81,82,83,84,85,86]. These counties are also the most health-disparate counties and have the lowest health outcome ranking when compared to other counties across the state. In 2022, the adult obesity prevalence rates in Wayne County and Macomb County were 38% and 31.9%, respectively [83]. In MI, 13.3% and 16.1% of children aged 2 to 4 are obese or overweight. In addition to these challenges, Metro Detroit is a low-healthy food access area [69], causing children in Macomb and Wayne Counties to experience food insecurity at rates of 13% and 18% versus the state rate of 15.9% and the national rate of 12% [83,84,85,86]. In 2019, 45.4% and 23.6% of residents in Wayne and Macomb Counties benefited from the Special Supplemental Women, Infants, and Children (WIC) program, and 58.4% and 40.6% received food assistance, the rates of which are significantly greater than those of the residents of other state counties [83,84,85,86].

Additionally, job opportunities in the auto industry, which has attracted new immigrants over the past century, have been the lifeblood of these counties. Despite the presence of the auto industry, however, Detroit has the highest poverty rate among large cities in the U.S., three times higher than the national poverty rate [87,88]. Almost 46.5% of Detroit residents live in poverty [87,88]. The decline of the auto industry over time has contributed to high poverty rates, especially among immigrant populations working in this industry [87,88]. Nearly 7.8% of Detroit’s population consists of Hispanics and Latinos. Southwest Detroit has been known as a Mexican town for many decades and is one of the main destinations for Hispanic and Latino immigrants. Within Mexican towns, several independently owned ethnic grocery stores have profoundly impacted the community’s health by offering a wide variety of culturally appropriate healthy foods [77]. The residents of Dearborn in Wayne County also experience a high rate of obesity, at nearly 40%, which is much higher than the obesity rate in Wayne County (38%) and MI (35%) [1,86,87,88,89,90]. Nearly 30% of families in Dearborn experience food insecurity, and 60% live in poverty. In addition, 84% of all Dearborn residents do not eat the recommended daily portions of fruits and vegetables [86,87,88,89,90]. In Hamtramck, the highest prevalence of obesity is 39%, and the prevalence of physical inactivity is 40.5%. Almost 11.2% and 13.4% of the city’s residents have coronary heart disease and type II diabetes, respectively [89]. Nearly 49.7% of Warren residents are obese, 41.1% are physically inactive, 16.6% have type II diabetes, and 17.5% live in poverty [88,89]. Furthermore, the MI Behavioral Risk Factor Surveillance System (MIBRFSS) (2013) reports that an estimated 25.4% of Michigan Arab immigrants have no healthcare coverage, which is significantly higher than 17.5% of all MI adults. Almost 25.3% of Arab adults reported not seeing their primary doctor within the past 12 months due to cost, which is remarkably higher than 15.5% of all MI adults [87,91]. The high burden of chronic diet-related disorders among immigrants in Metro Detroit is coupled with various social and economic determinants, including low health knowledge, the acculturation process, immigration trauma, and cultural shock, suggesting that young children within these families are at the greatest risk of a health crisis [87,88,89,90,91,92].

### 2.2. Sample

The primary target audience of this qualitative inquiry included immigrant parents of children aged 2 to 5 years old (N = 18) who were consumers in the 30 participating independently owned grocery stores of the Great Grocer Project. Initial recruitment occurred in the GGP stores via the completion of a customer survey, which included questions related to the presence of children aged 2 to 5 in the home and a space to include contact information if the participant wanted to be contacted for further inquiry (e.g., an interview) A subsample of surveyed customers (N = 57) responded yes to the interview invitation, and the lead author contacted them via email or phone. During initial contact, parents with no children ages 2–5 and who did not identify as first- or second-generation immigrants were excluded from the study. Thus, 18 eligible immigrant parents completed the interview for this study. The participating parents were asked to share insights and experiences through semi-structured one-on-one interviews. Interviews of up to 30 min were conducted via Zoom or phone. A Spanish- or Arabic-speaking certified interviewer interviewed participants whose primary language was Spanish or Arabic

### 2.3. Data Analysis

Interview narratives provided detailed insights from the immigrant parents, captured their voices regarding SSB marketing, and permitted participants’ experiences to be understood [83,84,85]. This study employed a constructivist inductive grounded theory method of analysis whereby the participants were positioned to co-construct theories and themes throughout the collection and analysis of the data [93,94,95,96]. In this study, the first author developed initial line-by-line coding and then focused on coding. Initial coding was used to explore each data group rather than employing preexisting code groups. Line-by-line coding assisted in obtaining an in-depth understanding of the target community’s concerns [93,94,95,96]. Additionally, initial coding enabled the researchers to find gaps in the data and helped to most closely reflect the opinions and perspectives of the participating parents [93,94,95,96]. The authors explored quotes from the immigrant parents’ responses during the interviews. Initial codes, such as child-targeted marketing tactics, were produced via line-by-line coding and included actions and processes derived from the immigrant parents and were reflected in their statements. The following comparative step was to identify repeated codes, exclude them, and sift through the acquired primary codes via focused coding. The focused coding resulted in more selective and abstract codes, such as toys and gifts for SSB purchases. The focused codes were compared with other codes, which guided the authors to classify the data more accurately and define three thematic categories and key constructs that answered this study’s aims and research questions [93,94,95,96]. Successive memo-writing also facilitated the coding and early data analysis processes [93,94,95,96]. Memos were kept while developing primary and secondary codes and categories to document the viewpoints, thoughts, and attitudes of participants shared during the interviews [93,94,95,96]. The first author analyzed the interview responses using ATLAS-TI version 8.4 software (ATLAS.ti Scientific Software Development GmbH, Berlin, Germany).

### 2.4. Procedure

This study acquired an amendment for the semi-structured interviews via submission to Wayne State University’s (WSU) Institutional Review Board (IRB), number 065117B3X. The researchers obtained verbal consent from the parents participating in the interviews. Participants for the interviews were recruited via a verbal, electronic, or phone call invitation based on their responses to the questions at the end of the consumer intercept surveys in the GGP stores. Parents were scheduled for interviews at their convenience. To maintain confidentiality, the collected data were de-identified before analysis and stored on encrypted, secure servers. After completing the interviews, to assist parents in recognizing SSBs versus unsweetened and healthier beverages for their young children, interviewed parents were provided with a fact sheet containing simple visual elements and a short description of what a sugary drink is, the different brands and names, and some examples of available SSBs at grocers. Participants’ parents were also provided with a $20 gift card as an incentive for their time and participation.

## 3. Results

### 3.1. Description of the Interviewed Parents and Community Demographics

This qualitative study was conducted with 18 immigrant parents of children between the ages of 2 and 5 years old who were consumers in the 30 independently owned grocery stores located in Hispanic and Latino enclaves in Southwest Detroit and Arab and Chaldean enclaves in North Central Detroit, Warren, Hamtramck, and Dearborn, MI. Table 1 demonstrates the demographics of the participants who provided narrative input in this study. All the participants identified as female (N = 18, 100%), and most were aged between 35 and 44 (N = 8). When conducting the interviews, the majority of participants reported that they had lived at their current address for two to three years (N = 7), and over half of them rented their house for money (N = 12). Over half of the interviewed parents immigrated to the U.S. more than 10 years ago (N = 13), and the economic indicator showed that they earned between USD 20,000 and USD 34,999 annually (N = 8). Most participants identified as Arab American/Middle Eastern/Chaldean (N = 5), and the minority identified as African American/Black (N = 1).

### 3.2. Narrative Descriptions of Interviewed Parents

The immigrant parents discussed their perspectives about healthy eating and the in-store aspects of SSB marketing that encourage them to purchase SSBs for their young children. They also illustrated promising strategies that could assist them in purchasing fewer SSBs in grocery stores. Three thematic categories generated from the parents’ contributions to this study are outlined in Table 2 and Table 3. These categories are (1) non-supportive grocery store environments; (2) acculturation to the American food environment (Table 2); and (3) strategies to support reduced SSB consumption in children, which have two subthemes and six categories. These subthemes and categories are (1) parents’ actions, including (a) seeking self-educating opportunities through available nutrition programs in their community and within grocers, (b) recognizing healthy foods and beverages, and (c) adopting a healthy diet for children, and (2) grocery stores’ actions, including (a) healthy food and beverage marketing, (b) in-store education and promotion, and (c) linking families to supplemental nutrition assistance programs, such as WIC, the Supplemental Nutrition Assistance Program (SNAP), and Double Up Food Bucks (DUFB) (Table 3). These thematic categories best represent the narratives of collective perspectives and answer the research questions of this study. Quotes from the narrative reports that best displayed the meanings of the thematic categories are selected from each of the participants’ narratives and discussed again below.

### 3.3. Non-Supportive Grocery Store Environments

The first thematic category, non-supportive grocery store environments, was discussed by each parent regarding the abundant availability of SSBs in the targeted grocery stores and the overwhelming marketing tactics used for these easy-access beverages. The challenges of healthy eating and healthy purchasing were also reflected in the narratives. Parents declared that their local grocery store environments do not frequently support practicing healthy nutritional habits. Collectively, all parents described the innovative tactics used by the food industry to introduce SSBs to consumers in grocery stores, specifically for products targeting children. Parents identified some aspects of SSB marketing within grocery store environments, including the purposeful placements, price, and promotion of SSBs, which catalyze the negative impacts of SSB marketing on children’s eating patterns. For example, three parents indicated the following:


*“Whenever I go grocery shopping, I see something new on beverages shelves or refrigerators. These new things, beautiful bottles in grocery stores, are not good. They make us ugly, but I guess my children are addicted to looking for new juice. It’s bad, but I have no time to fight.”*
(parent participant #3)


*“For purchasing organic or sugar-free beverages, there is not any signage or brochure within the stores. It’s nothing; instead, there are several shelves full of colorful and funny shape sugary beverages and the refrigerators near cashier full of them and you have to wait to pay and the best time for catching children’s eyes.”*
(parent participant #8)


*“We want to serve healthier beverages, but they become pricy when you have two or more children and I never see any off or sale on these products to buy. I think companies prefer them to be spoiled rather than reducing the price.”*
(parent participant #3)

Likewise, parents discussed the deceitful marketing practices in grocery store environments. Parents noted that they could obtain sugary beverages positioned at both children’s and adults’ eye levels. Parents believed that the food industry worked with local grocery stores to provide an abundance of SSBs and that food companies determined the placement of SSB products. Two of the mothers illustrated the connection between SSB placement and SSB purchases:


*“You know, children want to pick up whatever they see, and almost all the sugary beverages are at their height. If those drinks were in higher levels, they might have a little access to them. They are even at the entrance over the counters, everywhere. It only needs you to turn your head and grab one.”*
(parent participant #4)


*“It is so different when the kids come grocery shopping. Everything must be handed to them. There is no exception until you have half a shopping cart full of their requests; however, if the beverages won’t be at their sight, it is easier not to purchase unhealthy stuff. I cannot prevent them from looking around.”*
(parent participant #16)

The parents’ narratives depicted that despite families trying to learn about different healthy food and beverage items, grocery stores enable SSB marketing techniques to impact young children’s eating habits. Parents reported that mass marketing practices varied over time but often resulted in the high availability of SSBs within grocery stores. The parents’ narratives outlined that some SSBs are marketed as healthy and necessary options for growth. These beverages are then purchased by parents and served to young children. One parent’s narrative reported that they used to buy a type of SSB that they believed would boost the long-term health of the immune system. In the narrative reports of the parents, the authors observed that families were heavily exposed to SSB marketing. SSBs could be obtained in designated spots in grocery stores if parents did not have sufficient knowledge to shop for healthy foods. Approximately half of the parents reported that grocery stores should work with the community to promote and make readily available healthier foods and beverages. The parents’ narratives revealed that no information was provided to families about the availability of unsweetened beverages within the targeted grocery stores. Nor were marketing messages for those products provided in stores to improve children’s healthy eating behaviors. Regardless of the type of SSB marketing noted by parents, these tactics were reported to change children’s eating patterns and are supported by grocery stores. Parents also stated that they are continually concerned about their children’s health and nutritional behaviors. For example, despite the availability of unsweetened beverages in grocery stores, most parents requested more guidance on recognizing and finding them. The parents’ narratives exposed a perceived need for more healthy nutritional knowledge and skills.

### 3.4. Acculturation to the American Food Environment

The second thematic category, acculturation to the American food environment, was displayed in almost all narrative reports. The parents’ narratives of their lived experiences served as descriptors for this category. Participants also described religious, cultural, and linguistic factors that impacted their selection and purchase of beverages and foods. For instance, two parents stated the following:


*“We are vegetarian and try to obtain plant-based milk and other drinks for our children, but this store does not have them all the time, and we replace them with the other beverages that we can find here. It is hard for me to understand everything that is written on food boxes or juice bottles. I try to look and see what other people grab and buy them.”*
(parent participant #4)

*“We only purchase and consume Halal foods and drinks, and it is for all family, but sometimes it is hard to find what you regularly eat in this store, and you have children. We cannot find foods and beverages we used to eat, and we buy something else.”*.(parent participant #5)

Almost all parents explained their experiences with deteriorating eating habits after immigration due to limited access to healthy foods and beverages and increased consumption of less healthy American foods (e.g., fried foods, fast foods, and highly processed foods). Parents also discussed the link between their lifestyles after immigration and their nutritional behaviors, which connected the negative impacts of food and beverage marketing to worsening eating habits and diets post-immigration, specifically among their children. Additionally, parents declared that protecting children from exposure to all attractive advertisements is beyond their power. Below are some illustrative quotes from participating mothers:


*“I used to do more home-cooked meals and beverages, not so much sugary and oily. When I was back home, I had more free time to do this, but life has a hectic pace now. There is a difference between full of sugar and unsweetened juices a huge difference between organic milk and cool-aid or homemade lunch with ready to eat lunch. I would love my children to consume healthier and unsweetened beverages.”*
(parent participant #1)


*“Well, eating changed because of immigration. Those were huge changes because we ate and drink more local and homemade foods and beverages. We did not purchase many sugary drinks and pop back in our country. But here my children see different things on TV or school, and they want to eat them.”*
(parent participant #3)


*“There are different foods and drinks that we consumed, but immigration changed your access. We rely more on local stores in your neighborhood where you feel more comfortable purchasing familiar things or even trying new things. I try to ask staff and rely on whatever they offer in store and what I understand from them and the package. I hope what I buy will be healthy for my children.”*
(parent participant #12)


*“Many kids with their parents are within the grocery stores. You cannot control their eyes to not look around in the grocers. Everything is new and attractive to them, specifically when you come from another country.”*
(parent participant #4)

All parents reported that the changes in their communities’ nutritional environments due to immigration and the fact that SSB marketing was easy to observe and detect profoundly influenced the eating patterns of their families. Parents also reported that the constant marketing of unhealthy items in grocery stores employing different methods, such as eye-catching packaging, physical placements, and promotional offers for SSBs, made it difficult to establish healthy eating behaviors, which, in turn, caused health problems for their children. Parents stated that limited access to unsweetened beverages in parallel with innovative SSB marketing strategies had impacted their purchasing behaviors and their children’s nutritional behaviors. Quotes from the narrative reports are organized in Table 2.

### 3.5. Strategies to Support Reduced SSB Consumption in Children

The third thematic category, strategies to support reduced SSB consumption in children, was also discussed in almost all narrative reports. Grocery stores’ actions and families’ actions, the subthemes of this category, were prominent in narratives describing promising strategies to reduce SSB purchases and consumption. Parents correspondingly stated that the consistent increase in SSB marketing within grocery stores had caused negative changes in the healthy behaviors of their children. The parents’ narratives also explained how the impact of SSB mass marketing (on their children’s health) could be reduced by employing simple solutions (Table 3). For example, two parents reported in their narratives that they had noticed beverage sales stands at the entrances of some grocery stores and needed clarification about the beverages’ healthfulness. Parents thought the same marketing scheme could be used to introduce healthy unsweetened beverages or plant-based milk to attract consumers’ attention. As one parent mentioned, “I saw a stand marketing some juice, it was new to me, and I thought maybe I buy one to see what it is and how it tastes” (parent participant #2), and as another parent described, “So giant stand why it is not for the vegetables. For healthier shopping, we need to see healthier foods” (parent participant #18). Likewise, two other parents believed that limiting SSB marketing would positively influence their children’s requests to purchase different beverages, stating, “I have tried this when they do not see something, they will not ask for that” (parent participant #6) and “a little less advertisement will not decrease the benefits of the grocery stores. Why do they insist on having a different range of unhealthy beverages and pops?” (parent participant #8). Furthermore, approximately half of the parents illustrated the connection between limited SSB consumption, in-store healthy nutrition education, and the promotion of healthy beverages, which complemented the narrative of one parent, detailed as follows:


*“I guess information on healthy juices is important when grocery shopping for family, specifically children. If everybody knows healthy products, they can trust, purchase, and consume. If a grocery store doesn’t see it necessary to share nutritional product information, you know, not a nice food environment. It would help if you had a nice food environment to make the children healthy.”*
(parent participant #17)

Parents also discussed unreliable labeling and the need for in-store healthy marketing and education, stating, “I have seen bottles of juice labeled with Omega-3, but we were often not sure about it. We need more information” (parent participant #16), and “If I am supposed to give my children unsweetened beverages, I have to be able to see them physically, get information on them, and taste them within the stores” (parent participant #15). Another parent focused on in-store promotion and sales: “If I do not know healthy juice and I do not afford to buy them, how I am supposed to serve those healthy beverages to my three children?” (parent participant #14). Parents also sstressed the importance of having access to ethnic beverages and foods. Three parents suggested that the availability of ethnic beverages within local grocers could assist them in choosing healthier beverages that they knew well and used to consume in their home countries. As illustrated by one of the Arab parents, “local grocery stores maybe can offer a bigger selection of our traditional beverages, like date milk or ginger milk or honey milk or rosewater milk without sugar and preservatives” (parent participant #11). As another immigrant mother said, “we like to see and buy more traditional beverages in nearby stores” (parent participant #11).

Almost all parents explained that they needed simple tips and tactics to easily recognize healthy beverages. For instance, one mother mentioned that “Maybe juice should have some warning signs for sweet beverages. Then you even do not need to be good in English to find out what is better for your child” (parent participant #12). In addition to mentioning a need for healthy eating knowledge, parents reported that healthy marketing within grocers could also assist them in choosing and purchasing healthier beverages. Furthermore, parents acknowledged their low levels of trust in local grocery stores due to lower access to healthier foods and beverages that influence the health of their children and family. As one mother said, “I am uncomfortable grocery shopping at this store. They have few good things” (parent participant #17), and as another parent mentioned, “I cannot talk about what I need for my children. They do not have time to help to find what I need” (parent participant #14). Parents were questioned about how they could make the eating patterns of their young children healthier, and they disclosed their interest in collaborating with grocery store owners/managers to create more supportive eating environments. Parents discussed the positive impact of store owners’/managers’ efforts to implement strategies to convert grocery store environments into healthier ones. Parents also acknowledged that relationships between families and grocery store owners/managers could be strengthened to develop healthier eating environments. One parent explained the impact of creating healthy food environments for healthy purchases and healthy eating in the following remark: “Store owners of the local grocers can offer sales on healthy beverages and ask food companies to bring in a sample size of beverages to offer to families” (parent participant #13). Parents reported that the current grocery environment acts as a barrier to purchasing healthy foods and beverages. Parents also noted that more strategies could be implemented to create healthy grocery stores, such as in-store taste stations promoting healthy items, free samples of healthy beverages, bilingual brochures, and healthy product introductions at cashiers. Most participants reported that securing easy access to unsweetened and healthy beverages could create a reliable bond between families and grocery stores.

Parents also noticed that sales profits and the desire of grocery store owners/managers to prioritize financial gains were driving factors in overwhelming families with SSB marketing. The parents suggested that grocery store owners/managers should be fully aware of the consequences of SSB promotion and marketing on the community’s health. One parent reported that grocery store owners/managers needed to interact with them to learn which unsweetened beverages would be consumed in homes and therefore understand how to respond to the community’s demands and positively impact the community’s nutritional behaviors. Parents also noted that unsweetened beverages are unavailable or limited compared with SSBs at local grocers. The interviewed parents reported that they typically purchased and served high numbers of SSBs per grocery shop (Table 3). Overall, parents reported that SSB marketing within grocery store environments profoundly impacted the practice of healthy nutritional behaviors in young children. However, grocery store owners’/managers’ efforts to build dynamic relationships with community members to improve grocery store environments could be a promising strategy for community health promotion. Quotes from the narrative reports are organized in Table 3.

## 4. Discussion

This study found that non-supportive grocery store environments and acculturation to the American food environment were two main obstacles that immigrant parents face in providing an SSB-free diet to their young children. Parents reported that non-supportive grocery store environments provided communities with a high availability of SSBs at low prices. At the same time, the lack of in-store marketing or promotion for unsweetened beverages made practicing healthy eating difficult for parents. SSB marketing in grocery store environments also affected parents’ perceptions of the food environment and enhanced their awareness of the availability and affordability of SSBs. Parents’ nutritional perceptions combined with individual variables, including sociodemographic and psychosocial characteristics, increased the likelihood of them eventually purchasing SSB products and serving them to their young children. Additionally, how immigrant parents are acculturated to the American food environment impacted the parents’ decisions at the point of purchase [97,98]. Almost all participating immigrant parents explained how acculturation to the American food environment negatively influenced their food and beverage choices and eating behaviors. This is aligned with the findings of related studies that suggest that, regardless of the duration of residency in the U.S., immigrant families have challenges adapting to the American food environment. Those who are more acculturated to the American food environment purchase and consume more unhealthy foods and SSB products, which can lead to higher rates of overweight and obesity because the host country’s food culture is more obesogenic [97,98]. There have been documented associations between acculturation and obesity rates in Hispanic and Latino groups in the U.S. [97], where higher acculturation reduces the chance of maintaining a healthy weight among these populations [98,99,100,101,102,103]. Obesity and type II diabetes are also more prevalent among the most acculturated Mexican immigrants in the U.S., and the least acculturated Mexicans have lower rates of these disorders [103,104,105,106]. Likewise, children of U.S.-born and more acculturated mothers are more obese than the children of foreign-born and less acculturated mothers [103,104,105,106]. Furthermore, the age- and sex-adjusted prevalence of chronic diet-related disorders, such as obesity, is 8% among immigrants living in the U.S. for less than one year and doubles to 19% among those living in the U.S. for at least 15 years [107,108]. This illustrates that the longer an individual resides in the U.S., the higher the risk of developing diet-related disorders [107,108]. In addition to acculturation, socioeconomic status could also impact the adoption of health-risk behaviors, such as unhealthy eating behaviors, by immigrants within host countries [107,108].

The process of pairing the authors’ observations, experiences, and literature review with parents’ narratives provides a clear understanding of the need to develop healthier grocery store environments to enhance parents’ knowledge and awareness of healthy eating. Strategies to support reduced SSB purchases and consumption were frequently communicated in the parents’ narratives. Almost all parents showed strong motivation to improve and maintain their children’s healthy eating behaviors by enhancing their nutritional knowledge and collaborating with grocery stores. Parents mentioned that if they could be provided with healthy nutritional messages in grocery stores via healthy recipe cards or fact sheets, they could have more control over purchasing and serving healthy beverages to their children. However, even if all parent-led strategies were to be implemented, children would only be partially protected from the harmful impact of SSB marketing, and grocery store environments could still significantly influence their healthy eating practices. Parents referred to the role that grocery store managers/owners could have in reducing access to SSBs and making unsweetened beverages more available, accessible, and affordable for families. For example, parents discussed the need to place unsweetened beverages, especially those targeting children, in eye-catching spots in grocery stores and the provision of bilingual nutritional materials. However, the current grocery stores’ actions in promoting community health were minimal, if they occurred at all. Moreover, appreciation for the in-store promotion and sale of unsweetened beverages regardless of the financial profit or lost sales/products (e.g., due to expiration dates) was apparent in the parents’ narratives, and many parents indicated that their financial statuses made it difficult to pay for expensive beverages. Furthermore, after reviewing the narratives, it was apparent that most parents needed to be made aware of how to receive the benefits of federal nutrition assistance programs. The planning and implementation of healthy marketing in grocers that provides information on the requirements of federal supplemental nutrition programs while also providing parents with sufficient healthy nutritional knowledge within a supportive eating environment may help to mitigate the influence of SSB marketing on the development of early childhood obesity.

Grocery stores have enlisted choice architecture to maximize sales of products. For example, buying dairy products is one of the most common reasons for going to the grocery store, and they are usually placed at the back of the store so that shoppers have to travel longer through the store, increasing the chance for unplanned purchases [109,110,111,112,113]. A recent study in Metro Detroit showed that, in the grocery stores within the immigrant enclaves, there was a significantly higher availability of SSBs at lower prices than in grocery stores in less diverse areas of Metro Detroit. Signage featuring attractive elements for children was the most frequent in-store SSB marketing tactic used across these stores. Similarly, 56% of the immigrant enclaves’ stores had kids’ sugary yogurt drinks on the endcaps of the dairy section and checkout aisles, as well as non-100% juice and toddler sugary drinks. In almost 97% of the grocers within Metro Detroit, there were marketing materials/signs of cartoon characters near the milk/dairy/yogurt section [77,113]. Therefore, widespread SSB marketing toward toddlers within grocery stores in immigrant enclaves could be linked with the higher early childhood obesity prevalence among the immigrant population. Other studies have also indicated that the products located on endcaps or other free-standing displays account for 40% of all supermarket sales, and almost half of all supermarket sales of SSBs are selected in the checkout aisles [110,111,112]. Additionally, other research has shown that the obesity rate is lower in neighborhoods with healthier grocery environments (greater access and availability to healthy foods) [114]. Changes could be implemented within local grocers to improve grocery food environments and promote healthy eating behaviors through financial and non-financial interventions (e.g., shelf labeling, healthy eating posters, and in-store taste tests) that aim to reduce the consumption of SSBs among young children [115,116,117]. Recently, there has been increased attention on developing and implementing policy and environmental interventions to target SSB consumption, including taxation, which aim to improve community eating environments [118,119,120,121].

Finally, developing a supportive grocery store environment could provide families with a transparent approach regarding a grocery store’s accountability in promoting community health. Parents indicated that the more a store owner/manager prioritized consumers’ actual needs for healthier beverages, the more families saw themselves as being in health-supportive nutritional environments. Fostering a sense of commitment to community health in store owners/managers was revealed to be a vital component in developing healthy grocery stores and supportive food environments for communities from parents’ viewpoints, especially when most consumers belong to health-sensitive populations, such as immigrant families with young children. This concept has also been supported by the literature [122,123,124,125]. While all parents expressed a desire to make healthy nutrition decisions in grocery stores, complementing their abilities to act on those desires with in-store efforts could be more effective in sustaining behavioral change. This connection between store owners’ and parents’ efforts clarified the role of grocers as potential health-promoting agents to improve the health and wellness of children and the wider community. This study adds to the literature on early childhood obesity and highlights the connection between the impact of in-store SSB marketing and the nutritional behaviors of young children. This study’s results could empower immigrant families to enhance their control over their children’s health. In addition, the findings of this study could be disseminated to and implemented in other racial, ethnic/minority, and immigrant communities to assist with developing strategies for diminishing the impact of SSB marketing on early childhood obesity. Public health professionals could also employ the results of this research to advocate for policies that reduce the marketing of less-healthy beverages to young children.

### Limitations and Challenges

This qualitative study was limited by COVID-19 restrictions, which complicated direct interaction with parents. There were also a range of other limitations. First, selection bias may have occurred among the sample of interviewed parents, which could have impacted the findings; however, appropriate adjustments were made to include all the intended target audience members [94]. Immigrants and refugees are a hard-to-reach population, and the sampling in this study was limited by recent detention or deportation due to illegal residency status; therefore, some of the potential participants were hesitant to be contacted to avoid legal authorities [94,95,96], resulting in a small pool of 18 participants vs. the expected 25 immigrant parent interviews. Moreover, selection bias could have been caused if only parents who had already adopted healthy eating habits responded to interviews. The researchers tried to recruit parents who did not volunteer to participate in the interviews, ensuring that the selected subgroup reflected the target population regarding their key characteristics. Additionally, subject recruitment and interviews were conducted by CITI-trained bilingual interviewers, which aimed to reduce selection bias. The second limitation to acknowledge was researcher bias, as the authors had concurrent roles in the GGP and interacted with some administrators and support staff in the principal study [94,95,96]. Grocery store owners and staff, government officials, and medical professionals were not engaged in this study, yet they likely have influence over promoting healthier grocery store environments. Therefore, to address this limitation, responses to interview questions and parents’ narratives were carefully reviewed for alignment with research aims and with consideration of the one-sided approach. Additionally, the authors used member checking and an audit trail to reduce researcher bias and ensure that the data collection, data analysis methods, and population sampling aligned with the qualitative inquiries [94,95,96]. Peer debriefing was also a part of the process throughout different phases of the study, as the lead author discussed the research progress with her colleagues and assigned scholarly advisor to receive constructive feedback [94,95,96].

## 5. Conclusions

This study focused on understanding how SSB marketing tactics within independently owned grocery stores influence the purchase of SSBs among immigrant parents for their young children. This study also assessed strategies that could reduce SSB purchases and consumption from the viewpoint of immigrant parents. To the best of the authors’ knowledge, there were no other studies of this nature in the public or community health literature that outlined the objectives of this qualitative multiple-case study at the time of its completion.

The findings of this study indicate that most participating immigrant parents often expressed concern about the influence of grocery store environments on their children’s eating patterns. The parents frequently referred to grocery store owners/managers as potential community health sources who are capable of offering healthy foods, enhancing consumers’ healthy nutritional knowledge, and facilitating access to healthy foods and beverages; however, the current level of healthy eating support from grocery stores is extremely limited, if it occurs at all. Therefore, participating families could not assuredly rely on grocery stores as health-promoting resources. Participating immigrant parents believed that developing a supportive community food environment should be planned in partnership with consumer families. This collaboration could assist in meeting the immigrant families’ actual nutritional demands and could lead to more effective mitigation of the impacts of SSB marketing on nutrition-related behaviors in young children within these families.

The findings of this study could also be essential for designing successful community-based health promotion practices. As demonstrated in the results, there is widespread SSB marketing targeting toddlers in grocery stores, and even if immigrant families practice healthy nutritional behaviors, unsupportive healthy nutritional food settings and acculturation to the American food environment hinder the ability to obtain unsweetened beverages and maintain a healthy diet. Thus, independently owned grocery stores are key community partners that could provide families with adequate access to healthier options, promote children’s healthy eating behaviors, and act as health-promoting resources. Likewise, once parents are actively engaged with store owners/managers in converting grocery store environments to accommodate healthy eating environments, adopting an SSB-free diet could be more achievable. These strategies could help guide grocery stores to stock, promote, and sell unsweetened beverages, likely reducing SSB purchases and consumption and decreasing the prevalence of early childhood obesity in young children within immigrant communities of color.

## Figures and Tables

**Table 1 ijerph-20-06536-t001:** Demographics of 18 interviewed parents.

Race/Ethnicity	City of Residency	Years of Immigration	Education	Annual Income	Years of Living at the Current Address
Arab/Middle Eastern/Chaldean (N = 5)	Detroit (N = 7)	More than 10 years (N = 13)	Less than high school (N = 1)	Less than USD 20,000 (N = 5)	Less than 2 years (N = 6)
Latino/Hispanic (N = 4)	Dearborn (N = 4)	Less than 10 years (N = 5)	High school diploma (N = 4)	USD 20,000–USD 34,999 (8)	2 to 3 years (N = 7)
White (N = 3)	Warren (N = 4)		Some college; no degree (N = 5)	USD 35,000–USD 49,999 (5)	3 to 4 years (N = 2)
Asian (N = 3)	Hamtramck (N = 3)		Associate degree (N = 1)		More than 4 years (N = 3)
Bengali (N = 2)			Bachelor’s degree (N = 2)		
African American/Black (N = 1)			Master’s degree (N = 5)		

**Table 2 ijerph-20-06536-t002:** Narrative descriptors of thematic findings.

Identity Descriptor Categories	Parents		
Non-supportive grocery store environments	“Whenever I want to buy something healthier, I see no sale on that item, even for several months. They are expensive and not permanently available.”	“I am expecting new marketing whenever I enter the grocery markets. As soon as you enter, you are faced with a doll-shaped juice bottle, which even attracts me.”	“Children insist on buying it, and owner of the grocery stores will offer it more. It is a vicious cycle. I asked the store manager why you don’t sell unsweetened milk? He responded: Well we tried, and they spoiled, so weno more.”
Acculturation to the American food environment	“Even if I don’t want to buy for them (children). They know all the products from TV, classmates, grocery stores’ shelves. They want to try all the new tastes that they have never tried back in our country before.”	“I used to make more home-cooked meals and beverages, not so much sugary. When I was back home, I had more free time to do this, but life has a hectic pace now. There is a difference between juices full of sugar and unsweetened juices—a huge difference between organic milk and Kool-Aid. I would love my children to consume healthier and unsweetened beverages.”	“A lot of kids are with their parents within the grocery stores; we cannot find all the things we used to eat, and we have to buy new things in stores.”

**Table 3 ijerph-20-06536-t003:** Narrative descriptors of thematic findings.

Strategies to Support Reduced SSB Consumption in Children	Subthemes	Parents
Families’ actions	(a) Seeking self-education opportunities through available nutrition programs in their community and at the grocers; (b) Recognizing healthy foods and beverages; (c) Adopting a healthy diet.	“I spend money and sometimes the only things I have is whatever I see on shelves. I see no harm in hiring someone by grocery owners to introduce healthy beverages to customers or if cashiers can do that and have some sample there it would be great. I believe they can convince parents to try healthy ones and if not, at least they make efforts to spread the healthy words.” “If I see some information about foods in Spanish, I think it is familiar food. It is my feeling.”
		“The food marketing company can at least label sugary beverages with some simple warning label. Even if we do not know English, we can understand it is something that makes us fat and it may stop me to buy.” “If you see information in your language, would you go in and see what they are? Yeah, I would go and check it out.”
		“I would like to buy healthier beverages for my children, like organic almond milk or coconut milk, but they are too expensive for me to and also the regular milk are what stores in our neighborhood sells so I cannot afford organic ones, but I do try to encourage my kids to consume when they are on sale...I also try to encourage them to pick a healthier juice, of course if they are on sale and available.“ “I think if I try to be on a vegetarian diet, it is helpful to make my children familiar with specific tastes and my personal experiences showed me that whatever you serve to children, they prefer it more and more. The problem is I cannot find soy or almond milk in this grocery store all days. More availability is helpful.”
Grocery stores’ actions	(a) Healthy food and beverage marketing; (b) In-store sales; (c) Linking families to supplemental nutrition assistance programs, such as WIC, SNAP, and DUFB.	“Healthy beverages can be placed in cashier refrigerators and cashiers can share information with customers. You know with correct information or fact sheet or small stands in check out. Free samples or tasting also work.” “I wish I could find real healthy products in all refrigerators. I wish there was a board or stand to introduce one healthy drink per month for my children. I have never seen it in here. To me, the small information also is valid. I am a mom of two young children and this handy information can absolutely help me in having healthy beverages on my dinner table.”
		“Always I had to buy regular juice or milk for my children. I really want to feed them with better one, but my income is not enough for that. So, I trade off having sweet juice for seven days with having organic milk for only three days. We don’t see much sale on specific beverages and the rest are affordable always. It would be great if we can have more discount on healthy things.” “The store owners can add more healthy items for families or put some sales on healthier juice. They are expensive and I have three children, If I want to make them happy with healthier items, I need to spend a lot and I cannot afford it out of sale.”
		“I have never heard of this double program and never seen box or places for picking up the healthy recipes card.” “If even there is, I never seen a banner for double up food here.”

## Data Availability

The data presented in this study are available upon request from the corresponding author.

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
