# Peer review of "Community Solutions to Increase the Healthfulness of Grocery Stores: Perspectives of Immigrant Parents"

_ijerph, 2023, doi:10.3390/ijerph20156536_

Round 1

Reviewer 1 Report

Dear Authors, 

Thank you for preparing this interesting manuscript. It is an important topic of how food environments influence the choices of people coming from different food environments.  Please see below, a few suggestions. 

Introduction: It would be good to have some information on the migrant population. You only included some information in the methods.

There are several studies showing that migrants adopt the health risks of their new environment. Maybe reference some of these. 

Methods: Some more information on the actual recruitment would be useful. How many people replied to the invitations and how many ended up giving interviews? How were they chosen? 

Did you offer any incentives to the participants? 

Discussion: Again, if you could discuss the previous studies on migrants and health risks? There is a recent review on the topic. 

https://www.ssph-journal.org/articles/10.3389/ijph.2022.1604437/full

Author Response

This paper was revised based on the reviewers’ comments. I appreciate the feedback provided by the reviewers and feel the paper is strengthened by revisions made based on their concerns. The Reviewer 1 comments were addressed point by point below, and changes are trackable throughout the paper identifying revisions to the original version.

Reviewer 1

Thank you for your careful consideration and review of our paper. We have addressed your feedback point-by-point below.

Thank you for preparing this interesting manuscript. It is an important topic of how food environments influence the choices of people coming from different food environments.  Please see below a few suggestions.                                                                                                                               

Comment 1: Introduction: It would be good to have some information on the migrant population. You only included some information in the methods. There are several studies showing that migrants adopt the health risks of their new environment. Maybe reference some of these.                                                           Response: Thank you for providing a review and feedback on our paper. The introduction was revised for more clarity around the migrant population and contributors to the health risk adoption in host countries. New references and suggested reference were used to support the revised part of the introduction. Please see lines 101-104 and 132-138.

Comment 2: Methods: Some more information on the actual recruitment would be useful. How many people replied to the invitations and how many ended up giving interviews? How were they chosen?                                                      Response: Initial recruitment occurred in the GGP stores via the completion of a customer survey which included questions related to the presence of children aged 2 to 5 in the home and a space to include contact information if the participant wanted to be contacted for further inquiry (e.g., an interview). A subsample of surveyed customers (N=57) responded yes to the interview invitation, and the lead author contacted them via email or phone. During initial contact, parents with no children ages 2-5 years old and who did not identify as first or second-generation immigrants were excluded from the study. Thus, 18 eligible immigrant parents completed the interview for this study. We added details of recruitment to the paper for more transparency. Please see lines 238-246.

Comment 3: Did you offer any incentives to the participants? At the end of the interview sessions.                                                                                            Response: Participants were reimbursed with a $20 gas gift card as an incentive for their time and participation. We added this to the paper. Please see lines 289-290.

Comment 4: Discussion: Again, if you could discuss the previous studies on migrants and health risks? There is a recent review on the topic.                             Response: The discussion was revised for more clarity around the migrants and contributors to health risks adoption, such as residency duration and how this study aims to explicate and understand this concept. New references, including the suggested one, were used to support and make the discussion strengthen. Please see lines 554-560.

Reviewer 2 Report

The manuscript "Community Solutions to Increase Healthfulness of Grocery Stores: Perspectives of Immigrant Parents" presents a clear, thoughtful assessment of parents' perceptions of the challenges related to sugar-sweetened beverages (linked to high rates of obesity). 

The introduction clearly lays out the growing association between SSB and obesity rates in adults and children. The methods are well laid out and clear. The results are clearly presented, although Table 1 demographics is presented in an unusual way. Table 2 might be better positioned after the more detailed presentation of the findings.

The discussion could offer more in terms of connecting this study's findings to the work of others and expanding the implications. The study did not offer any analysis of the grocery stores themselves, but many other studies have. As grocery stores aim to sell products, could the authors present studies that have analyzed how products are placed within grocery stores in order to sell more. Specifically, the observations made by parents that stores place certain items at child level on purpose has been well documented in the literature. In addition, placing essentials on the perimeter of the store requiring customers to fully enter the store and be attracted to products they likely do not need. In addition, this study included 30 independently owned grocery stores. How might this affect the findings as these are likely stores with limited offerings and potentially at higher prices than larger chain grocery stores?

Also, I am curious about parents' statements about the high cost of healthier beverages. What about water consumption? There may be concerns about water quality in Detroit, but bottled water is cheaper than any other sold beverage. Was there any discussion about water and whether this isn't perhaps considered a safe beverage option? 

Along these lines, interviews explored attitudes about eating healthy, but what about understandings of what is and isn't healthy? There was a statement about plant-based milk, but these are often sweetened, and consequently contain more sugar than milk. I understand that parents felt that store owners should be providing more information, but what about recognizing store owners are in the business of selling products, not offering public service announcements? What about the role of government in providing information or even regulating product labeling or product placement in grocery stores? Or about the role of other information sources such as medical providers, or the expectations in the US for self-management?

Author Response

This paper was revised based on the reviewers’ comments. I appreciate the feedback provided by the reviewers and feel the paper is strengthened by revisions made based on their concerns. The Reviewer 2 comments were addressed point by point below, and changes are trackable throughout the paper identifying revisions to the original version.

Reviewer 2

Thank you for your careful consideration and review of our paper. We have addressed your feedback point-by-point below.

Comment 1: The introduction clearly lays out the growing association between SSB and obesity rates in adults and children. The methods are well laid out and clear. The results are clearly presented, although Table 1 demographics is presented in an unusual way. Table 2 might be better positioned after the more detailed presentation of the findings.                                                                            Response: Thank you for providing a review and feedback on our paper to make Table 1 more transparent, we added 18 to the table topic, which shows the total number of interviewed parents. Also, we added the letter “N” in front of the numbers within the table that indicate the number of cases in each category. All the numbers are out of a total of 18. Please see line 306 and Table 1. We also positioned tables 2 and 3 after the related descriptions to make more connections between the tables' details and findings explanations. Please see line 435 for table 2 and line 526 for table 3.

Comment 2: The discussion could offer more in terms of connecting this study's findings to the work of others and expanding the implications. The study did not offer any analysis of the grocery stores themselves, but many other studies have. As grocery stores aim to sell products, could the authors present studies that have analyzed how products are placed within grocery stores in order to sell more. Specifically, the observations made by parents that stores place certain items at child level on purpose has been well documented in the literature. In addition, placing essentials on the perimeter of the store requiring customers to fully enter the store and be attracted to products they likely do not need. In addition, this study included 30 independently owned grocery stores. How might this affect the findings as these are likely stores with limited offerings and potentially at higher prices than larger chain grocery stores?                  Respond: The discussion was revised for more clarity around connecting this study's findings to the other studies about specific placement and promotion of products within the grocery stores and how this study aims to explicate and understand this concept. We added new references to support the revised part, strengthened the discussion, and expanded the implications. Please see lines 589-614.

Additionally, Participant grocery stores in this study were all large format, full-service grocery stores with abundant food offerings. Given that the stores are independent vs. corporately owned and located in an urban environment, some food products may sell at higher prices in their stores than competitors (e.g., Wal-Mart) (Dombrowski et al., 2022; Dastgerdizad et al., 2023). Please see lines 35 and 185-189.

Comment 3: Also, I am curious about parents' statements about the high cost of healthier beverages. What about water consumption? There may be concerns about water quality in Detroit, but bottled water is cheaper than any other sold beverage. Was there any discussion about water and whether this isn't perhaps considered a safe beverage option?                                                                                Response: Although there is a high mistress in water consumption in Michigan after the Flint Crisis, in this study, we focused on understanding the experiences, opinions, and attitudes of immigrant parents of young children regarding why and how in-store marketing of SSBs can encourage them to purchase SSBs instead of healthier beverages such as plain water. We also looked for the strategies these parents believed could help them purchase fewer SSBs. However, why these families did not choose plain bottled water instead of SSBs was understood by focusing on previous studies and understanding the impacts of in-store SSBs marketing tactics on parents' purchasing behavior and children's eating patterns and preferences through obtained narratives in this study, which led to three thematic categories. In this study, parents were asked to answer one question regarding the type of beverages their children consume weekly. The question was, "Which of these beverages are consumed within your household within a given week? How often are these beverages consumed within a given week? By whom? Probe for Children". The list of beverages offered to parents to choose from them contained plain water and other safe and unsweetened beverages, such as 100% fruit juice, plain milk, and sugary drinks. Almost all the parents chose plain water as the least consumed beverage by their children. The reason for this can be the individual's perception of the safety of the water that influences whether they consume it or choose an alternative, such as SSBs (Pierce & Gonzalez, 2016; Yang & Faust, 2019). Robust evidence indicates that the perception of water quality is most influenced by belonging to racial or ethnic minorities and foreign-born groups (Pierce & Gonzalez, 2016; Yang & Faust, 2019), and lower-income and minority groups consumed relatively little plain drinking water. For example, there is significant mistrust among low-income African American and Latino-American families regarding access to safe drinking water (Pierce & Gonzalez, 2016). Therefore, water mistrust could still be a reason for a lower water consumption rate in states such as Michigan, especially after the Flint Water Crisis. Even after tap water was deemed safe, there has still been fear of drinking water due to lead contamination and increased preferences for drinking other alternatives, including SSBs (Hanna-Attisha et al., 2016; Patel & Schmidt, 2017). We added a paragraph regarding this point to the introduction, used additional references to support the topic, and made the discussion more understandable for the audience. Please see lines 118-125.

Comment 4: Along these lines, interviews explored attitudes about eating healthy, but what about understanding what is and isn't healthy? There was a statement about plant-based milk, but these are often sweetened, and consequently contain more sugar than milk. I understand that parents felt that store owners should be providing more information, but what about recognizing store owners are in the business of selling products, not offering public service announcements? What about the role of government in providing information or even regulating product labeling or product placement in grocery stores? Or about the role of other information sources such as medical providers, or the expectations in the US for self-management?                                                                                                                   Response: The main purpose of this paper was to find out the strategies from the viewpoints of immigrant parents that they believe can assist them in purchasing fewer SSBs within the grocery store environments. Even though healthy nutritional education has been proven to be an essential contributor to improving eating behaviors and diet patterns, the focus of this paper was not on that; however, after completing the interviews, to assist parents in recognizing SSBs versus unsweetened and healthier beverages for their young children, interviewed parents were provided with a fact sheet containing simple visual elements and a short description of what a sugary drink is, the different brands and names, and some examples of available SSBs at grocers. Additionally, this study is part of a more extensive and ongoing healthy nutrition intervention, including healthy food marketing within the grocery stores of Metro Detroit. For the healthy eating intervention part, a goal is to make targeted grocery stores health-promoting grocers, and the results will be analyzed and discussed in future papers. In this paper, we only mentioned providing a fact sheet to interviewed parents, but we revised that part of the paper and added more details to clarify it. Please see lines 285-289.                                                              One of the limitations of this study is that the grocery store owners and staff, government officials, and medical professionals were not engaged in this study. Yet, they likely have influence over promoting healthier grocery store environments. Therefore, to address this limitation, responses to interview questions and parents' narratives were carefully reviewed for alignment with research aims and with consideration of the one-sided approach. We revised the limitation part of the paper and explained this limitation in more detail. Please see lines 655-659.
